# Combination of *Bacillus velezensis* RC218 and Chitosan to Control Fusarium Head Blight on Bread and Durum Wheat under Greenhouse and Field Conditions

**DOI:** 10.3390/toxins14070499

**Published:** 2022-07-18

**Authors:** Juan Palazzini, Agustina Reynoso, Nadia Yerkovich, Vanessa Zachetti, María Ramírez, Sofía Chulze

**Affiliations:** Research Institute on Mycology and Mycotoxicology (IMICO), National Research Council from Argentina (CONICET), National University of Rio Cuarto (UNRC), Ruta Nacional 36 Km. 601, Río Cuarto X5804BYA, Córdoba, Argentina; reynosoa@exa.unrc.edu.ar (A.R.); nyerkovich@exa.unrc.edu.ar (N.Y.); vzachetti@exa.unrc.edu.ar (V.Z.); mramirez@exa.unrc.edu.ar (M.R.); schulze@exa.unrc.edu.ar (S.C.)

**Keywords:** Fusarium head blight, biocontrol, chitosan, wheat, deoxynivalenol reduction

## Abstract

*Fusarium graminearum sensu stricto* is, worldwide, the main causal agent of Fusarium head blight in small cereal crops such as wheat, barley, and oat. The pathogen causes not only reductions in yield and grain quality but also contamination with type-B trichothecenes such as deoxynivalenol. Prevention strategies include the use of less susceptible cultivars through breeding programs, cultural practices, crop rotation, fungicide application, or a combination of them through an integrated pest management. Additionally, the use of more eco-friendly strategies by the evaluation of microorganisms and natural products is increasing. The effect of combining *Bacillus velezensis* RC218 and chitosan on Fusarium Head Blight (FHB) and deoxynivalenol accumulation under greenhouse and field conditions in bread and durum wheat was evaluated. Under greenhouse conditions, both *B. velezensis* RC218 and chitosan (0.1%) demonstrated FHB control, diminishing the severity by 38 and 27%, respectively, while the combined treatment resulted in an increased reduction of 54% on bread wheat. Field trials on bread wheat showed a biocontrol reduction in FHB by 18 to 53%, and chitosan was effective only during the first year (48% reduction); surprisingly, the combination of these active principles allowed the control of FHB disease severity by 39 and 36.7% during the two harvest seasons evaluated (2017/18, 2018/19). On durum wheat, the combined treatment showed a 54.3% disease severity reduction. A reduction in DON accumulation in harvested grains was observed for either bacteria, chitosan, or their combination, with reductions of 50.3, 68, and 64.5%, respectively, versus the control.

## 1. Introduction

Wheat (*Triticum aestivum*) is one of the most cultivated cereals due to its nutritional contributions such as proteins, carbohydrates, minerals, B-group vitamins, and dietary fiber [1]. Worldwide wheat production has increased through the years and, for 2022, could reach 776.6 mt [2]. However, the changing global environmental conditions such as unexpected biotic and abiotic factors generate considerable yield losses and affect the grain quality. The main climatic factors that affect crops are temperature and rainfall, while biotic factors include weeds, insects, and a wide range of pathogens such as the *Fusarium graminearum* species complex (FGSC) [3]. *Fusarium graminearum ss* is the most destructive hemi-biotrophic pathogen on wheat, which mainly affects regions with warm and wet climates, especially during the anthesis stage. Under these conditions, water-splashed and wind-discharged macroconidia from sporodochia or ascospores from perithecia fall down on the host and start to colonize its tissues [4], causing blight on spikes and reducing grain formation or filling. During the infection process, species within the *Fusarium graminearum* complex have the ability to produce mycotoxins such as deoxynivalenol (DON) and nivalenol (NIV), as well as zearalenone (ZEA) and moniliformin (MON), all of which showed toxicity on humans and animals [5]. Maximum contamination acceptable levels for DON in cereal-based food were set by the European Commission, where these limits were fixed at 1250 µg Kg^−1^ in unprocessed common wheat and 1750 µg Kg^−1^ in unprocessed durum wheat for human consumption in the European Union [6]. In Latin America, Brazil established a maximum limit for DON in wheat flour and bakery products of 750 µg Kg^−1^ and 1000 µg Kg^−1^ for wheat bran, respectively [7]; Uruguay established maximum DON levels of 1000 µg/kg in wheat flour and crackers [8]. In Argentina, during the last 50 years, several FHB epidemics have occurred, causing several yield losses (up to 70%) and reductions in grain quality due to mycotoxin contamination [9,10]. Based on this, Argentina has also set-up maximum levels of DON for wheat flour of 200 μg Kg^−1^ for babies and children and 1000 μg Kg^−1^ for wheat-based products for general consumption [11].

The mycotoxins are of increasing worldwide concern and require efforts by different actors in the food and feed chains to look for strategies to prevent *Fusarium* species and mycotoxin contamination in the food and feed chains. Some strategies have included cultural practices, and chemical and biological controls [12]. The most effective management of *Fusarium* species includes chemical control, crop rotation with non-host species, conventional tillage burying the crop residues, and resistant cultivars [13,14]. Fungicides from azole families are effective for *Fusarium* species control and their mycotoxins, generally with a broad spectrum and long stability [15]. Despite this, the continuous (sometimes incorrect) application of fungicides can lead to the development of resistance of pathogens, a shift in species occurrence with increased aggressiveness (chemotypes), as well as the generation of environmental contaminations and risks for human health [16,17]. These have led to the reconsideration of new tools to control diseases such as the application of microbial antagonistic agents and natural antifungal compounds. Biocontrol using bacterial and fungal species with different mechanisms of action to control pathogens, such as competition for nutrients, production of inhibitory molecules, and induction of systemic resistance in host plants, have been evaluated [18]. Some antagonistic species that have been extensively studied are *Bacillus* spp. and *Pseudomonas* spp. [19], as well as *Trichoderma* spp. and *Clonostachys rosea* [20]. These microorganisms showed the ability to reduce either the level of *Fusarium* infection on several crops or mycotoxins accumulation [21,22]. Natural fungicides are also promising substitutes as chemical tools due to their antimicrobial activity, nontoxicity, biocompatibility, and biodegradability [23]. Chitin is one of the most important polysaccharides and can be easily found in a lot of organisms such as insects, lobsters, shrimp, and crabs [24]. It is also the major source of chitosan, a polymer semi-synthetic obtained through a chemical process denominated as deacetylation. It has been demonstrated to inhibit microorganisms such as fungi, bacteria, and viruses; additionally, it can control plant diseases by disrupting the cell membrane and preventing DNA/RNA synthesis, causing cell death [25,26]. In addition, it has the ability to induce resistance in the host plants and enhance the rhizosphere biodiversity [27]. Regarding *Fusarium* sp. pathosystems, studies carried out by our group during the last two decades showed the efficacy of these eco-friendly alternatives to control plant diseases. For example, several bacteria were isolated from wheat anthers and proved to control *F. graminearum* growth in vitro and DON production on irradiated wheat grains; later, FHB disease under greenhouse trials was controlled by some strains [28]. Further studies revealed betaine accumulation on vegetative cells of *B. subtilis* RC218 (now *B. velezensis*), which resulted in a higher biocontrol activity against FHB under greenhouse trials [29], also demonstrating plant defense activation by salicylic and jasmonic production in the plant [30]. Genome sequencing allowed us to locate the strain in the *B. velezensis* clade and to identify several secondary metabolite clusters with known antifungal activity and the production of Ericin S, a L-antibiotic that was first described for this species [19]. Finally, field trials on bread and durum wheat revealed the effectiveness of *Bacillus velezensis* RC218 and *Streptomyces albidoflavus* RC87B in reducing FHB disease severity on heads, and DON accumulation and *F. graminearum* DNA on harvested grains [19,30]. Under greenhouse or field trials, chitosan was only individually evaluated against FHB, with only one study using chitosan or *Pseudomonas* sp. as effective treatments against *F. graminearum* [31]. Based on previous data, the aims of this study were to evaluate the effect of chitosan and *Bacillus velezensis* RC218 alone and in combination to reduce Fusarium head blight parameters and DON accumulation on wheat under greenhouse and field trials. 

## 2. Results

### 2.1. Greenhouse Evaluation

*Bacillus velezensis* RC218 and chitosan were evaluated separately and in combination in order to control FHB as the bacterial strain demonstrated good biocontrol activity in previous studies [19,28,30]. Both *B. velezensis* RC218 and chitosan (0.1%) demonstrated control of FHB, diminishing disease severity by 38 and 27%, respectively, although not significantly. Surprisingly, the combined treatment resulted in a statistically significant reduction in FHB severity by 54% according to Duncan’ *a posteriori* test (*p* ≤ 0.05) (Figure 1), but an additive or synergistic effect of the combined treatments could not be evidenced (Ee = 54.64, Er = 54.11). A negative control (water) was also evaluated, with no disease symptoms.

### 2.2. Field Trials on Bread and Durum Wheat

*Bacillus velezensis* RC218 and chitosan treatments were applied at the anthesis period during 2017/18 and 2018/19 bread wheat harvest seasons and the 2017/18 durum wheat harvest season. For bread wheat trials, climatic conditions did not help in the disease development despite the utilization of fine misting during the critical period (flowering), conducive to low disease incidence but allowing evaluation of the different treatments applied; disease severity was higher during the first trial. Figure 2 shows the combined bread wheat trials for 2017/18 and 2018/19. Pathogen treatment showed a 20.8% disease severity during the first trial and 4.9% during the second. Biocontrol treatment was able to control FHB severity during both trials, with a reduction of 53 (statistically significant) and 18% for 2017/18 and 2018/19 trials, respectively. Chitosan treatment statistically reduced FHB severity during the first year by 48%, but no differences were observed for the 2018/19 bread wheat trial in comparison with the *F. graminearum ss* treatment. The combination of *B. velezensis* RC218 and chitosan resulted in a significant reduction in disease severity of 39% during the first trial and a 36.7% reduction in the second trial. At harvest, FDK was determined on each treatment. As no differences were observed between both trials, data were pooled. Observations revealed that both bacteria and chitosan reduced FDK, and the combined treatment was only successful on bread wheat (Table 1). The deoxynivalenol content on grains was determined on harvested grains. A reduction in DON accumulation on grains was observed for both bacterial, chitosan, and their combination, with reductions of 50.3, 68, and 64.5%, respectively, in comparison with the pathogen treatment (Table 1). 

### 2.3. Field Trial on Durum Wheat: Disease Evaluation, DON Content, and Pathogen DNA Quantification on Harvested Grains

The 2017/18 durum wheat trial showed an average disease incidence of 38%, with no effects on reduction by the different treatments applied (data not shown). Treatment with the pathogen alone scored a 55.6% disease severity, and biocontrol and chitosan treatments showed statistical reductions. The highest reduction was observed for *B. velezensis* RC218 alone, with 58.6%, followed by the combined treatment and finally chitosan alone, with 54.3 and 49.6% of FHB severity, respectively (Figure 3). *Fusarium graminearum* DNA on grains from the pathogen treatment scored 92 pg mg^−1^; no significant reduction in DNA content was observed for biocontrol treatment (43.6%) and the combination with chitosan (28.2%), but not the natural compound alone. No correlation could be established between the DNA content on harvested grains and the severity observed in the field trial in the different treatments evaluated.

## 3. Discussion

One of the actual goals of agriculture is to produce healthier food and, at the same time, to develop cropping systems with a reduced impact on the environment and health [32]. Conventional crop production systems include mainly the use of agrochemicals to control pathogens in combination with other pest management strategies such as breeding programs, crop rotation, and cultural practices. For FHB, resistance is quite difficult to achieve as several traits (QTLs) have been associated but are still not providing full FHB protection [33]. New strategies for pest control have been developed in the last two decades, such as eco-friendly alternatives with a low impact on the environment, which include the use of natural products or microorganisms to control fungal pathogens [31,32]. Additionally, the increasing use of biologicals in organic farming to control pathogens has attracted consumers to “go for these more natural products” [34], also promoted by government in some cases as “biopreferred” [35,36].

The present study shows for the first time the utilization of a proved biocontrol agent, *B. velezensis* RC218, in combination with chitosan to reduce FHB disease severity and DON accumulation on greenhouse and field trials on bread and durum wheat. In previous studies, we demonstrated the effectiveness of a bioformulated product based on *B. velezensis* RC218 on the reduction in *F. graminearum* growth parameters in vitro and DON production on irradiated wheat grains and further evaluated under a greenhouse [28]; lately, field experiments on bread and durum wheat validated the biocontrol activity of the strain [19,30]. Our research group also evaluated the effect of chitosan on growth and mycotoxin production on the main wheat and maize *Fusarium* pathogens [37] and the possibility of a combined strategy. There are many studies conducted on the control of *F. graminearum* growth, DON production, or FHB disease on wheat by the use of biocontrol agents [38,39]. Endospore-forming bacilli are among the most widely biological agents used as biopesticides [40]. *Bacillus velezensis* RC218 was demonstrated to produce a wide range of antimicrobial compounds such as cyclic lipopeptides from iturin, fengycin, and surfactin families, with known activity against fungal pathogens [19], as it was also described for other *B. velezensis* strains isolated in China to control FHB [40]. There are also many studies (but fewer than biologicals) on the control of the pathogen by the use of chitosan or some derivatives [25,26], but the combination of biocontrol agents and chitosan has not been deeply well evaluated yet. For example, a greenhouse and field study on wheat and barley was carried out by evaluating *Pseudomonas fluorescens* MKB58 and a biochemical chitosan [31]. The commercial crab-shell-derived chitosan reduced FHB severity by more than 74% and *P. fluorescens* MKB158 by 48%, and the DON content was also reduced by these treatments, but the combination of them was not evaluated. Another study at the greenhouse level revealed that chitosan at 1000 ppm caused very low disease symptoms, which is in agreement with the present study using the same chitosan concentration [41]. 

On bread and durum wheat, both *B. velezensis* RC218 and chitosan were effective in reducing FHB disease severity when applied individually, where the combination of these treatments did not result in a better effectiveness in the control of the disease severity and DON accumulation. The observed results could be explained based on the low dose of chitosan used in the experiments as the inoculum dose used for *B. velezensis* RC218 has been previously demonstrated to be effective under field trials [19,30]. A study using chitosan hydrochloride at 0.5% applied on flag leaves and flowering heads on durum wheat showed a reduced FHB disease severity up to 70% [42]; when compared with our results on bread and durum wheat, chitosan only achieved a maximum of 49.5% of disease severity reduction, although the chitosan dose applied in our experiments was lower (0.1%). The concentration of chitosan tested in the present study on wheat spikes in combination with *B. velezensis* RC218 was based on a previous study performed by Zachetti et al. [37]. They showed under in vitro experiments that 1 mg g^−1^ was enough to reduce the growth rate of *F. graminearum ss,* and a DON accumulation reduction by 93–100% was achieved using chitosan at 0.5–2 mg g^−1^. Under field conditions, we obtained a 67–70% DON reduction by using chitosan at 0.1%, and this effectiveness was not observed by Francesconi et al. [42] when applying chitosan at 0.5% in durum wheat. Chitosan applied at flowering on a susceptible wheat cultivar was able not only to prevent FHB development but also increase the average grain weight per head, the number of grains per head, and the 1000-grain weight compared to the controls sprayed with water [43]. In our study, chitosan and biocontrol combination rendered a 20–40% grain protection in the spike by FDK index, but unfortunately, no additive or synergistic effect was observed. At root level, a combination of chitosan preparations with *Bacillus subtilis* formulations was also effective in increasing plant growing parameters, yields, and protection against root pathogens such as *Helminthosporium* [44]. Many studies have highlighted the effect of chitosan and/or its derivatives on plant growth parameters and their effect on eliciting plant defenses against *Fusarium* pathogens in wheat [27,42]. Additionally, some studies go deeper by evaluating the activation of defense pathways or triggering new metabolites production incited by chitosan-related metabolomic studies [43]. Other studies have also evaluated the impact of using chitosan and its derivatives on the main cereal crops in the era of climate change [25].

There are several studies combining biocontrol strains and chitosan on diverse pathosystems [45,46,47]. On the *Fusarium*-wheat pathosystem, Khan et al. [48] reported the activity of several *Pseudomonas* strains combined with chitosan in preventing seedling blight caused by *F. culmorum* on wheat and barley; additionally, the reduction in the *Tri5* gene expression (trichothecene pathway) was observed by the application of the treatments in the stem base of wheat plants. Regarding *F. graminearum* on wheat, chitosan and *P. aeruginosa* were only individually evaluated on FHB disease parameters reduction [31]. Differences observed in relation with disease severity levels and DON accumulation by the treatments evaluated during bread wheat harvest seasons could be related to the effect of unfavorable climatic conditions for FHB occurring during 2017/18.

To the extent of our knowledge, this is the first report on the effectiveness of a proven biocontrol agent (*B. velezensis* RC218) in combination with chitosan to reduce FHB disease parameters and DON accumulation in both bread and durum wheat under greenhouse and field trials. Despite the combined treatment at the greenhouse showed a disease reduction similar to the sum of treatments applied alone, an additive or synergistic effect was not possible to evidence statistically; thus, further research needs to be conducted in order to optimize the chitosan dose and evaluate the possible (desired) additive/synergistic effect to be applied under either greenhouse or field trials.

## 4. Materials and Methods

### 4.1. Biocontrol, Chitosan, and Pathogens’ Preparation

*Bacillus velezensis* RC218 used in the present study was originally isolated from wheat spikes and successfully evaluated under greenhouse and field conditions in several harvest seasons [19,30]. A bioformulated product was previously obtained from freeze-dried cells [30]. Briefly, bacteria was produced in basic medium supplemented with NaCl to a final water activity of 0.97 to induce betaine accumulation [29]. After 48 h at 28 °C, the biomass was centrifuged and resuspended in distilled water with the addition of a protectant before freeze-drying. One gram of freeze-dried cells was resuspended in sterile distilled water plus tween 80 (0.05%), serially diluted in peptone water, and colony-forming units (cfu) were determined by surface plate counting. The bacterial inoculum was adjusted to 1 × 10^6^ cfu mL^−1^ for all the performed experiments. Low-viscosity chitosan was purchased from Sigma-Aldrich (CAS 9012-76-4). Ten grams per liter of chitosan was dissolved in 1% acetic acid, stirred for 24 h at room temperature until complete dissolution, adjusted to pH 5.6 with NaOH, and autoclaved. This solution served as a stock to prepare 0.1% working solutions for the experiments. *Fusarium graminearum* strains used in the experiments belong to the Research Institute on Mycology and Mycotoxicology (IMICO) collection (Río Cuarto city, Argentina). For greenhouse and field experiments on bread wheat, *F. graminearum ss* PER5612 and MJ1112 were used; meanwhile, *F. graminearum ss* S-5 and S-17 were evaluated in the durum wheat field experiment. *F. graminearum ss* strains used on either bread or durum wheat were specifically isolated as pathogens of the specific cultivar. Either for greenhouse or field experiments, pathogen inoculum was produced in Mung bean broth and applied at the anthesis period, with 50% of the spikes at the flowering stage (Feekes stage 10.5.2–10.5.3) [49]. The *Fusarium* macroconidia concentration was adjusted in an hemocytometer chamber to 1 × 10^5^ conidia mL^−1^. 

### 4.2. Greenhouse Evaluation

A susceptible-to-FHB Bread wheat, cultivar BioINTA 1005, was used in the 2016 harvest season to evaluate the possible combination of the biocontrol and chitosan. Eight wheat seedlings in a 10 L pot (containing soil, peat, and perlite; 70:20:10, respectively) were grown for 12 weeks under a greenhouse, with controlled temperature (24 °C) and a natural photoperiod length. Plants were watered and fertilized as needed. At the anthesis period, 10 mL of each treatment (Table 2) was applied on the heads of each pot with manual sprayers located at 20 cm and 45° over the heads. The experiment consisted of three pots per treatment in a randomized block design. Inoculated heads were maintained at 90–100% humidity with automatic foggers for three days to ensure pathogen colonization. After 16 days from the treatment´s application, FHB disease was evaluated on heads by comparing with a 0–100% disease scale proposed by Stack and McMullen [50]. In order to determine an additive or synergistic effect, interactions between treatments were analyzed with the Limpel´s formula proposed by Richer D. [51], using the equation: Ee = X + Y − XY/100, where X and Y represent disease reduction of treatment X (*B. velezensis* RC218) and Y (chitosan 0.1%), respectively. Ee is the expected disease reduction of the combined treatment. If the observed effect of the combined treatments (Ereal) is greater than the expected effect (Ee), then synergism is said to be positive. Similar values (Ee vs. Ereal) are thought to be only an additive effect and lower observed values versus expected ones indicate an antagonism of the treatments.

### 4.3. Field Trials on Bread and Durum Wheat

#### 4.3.1. Experiment’s Locations, Design, and Disease Parameters Evaluated

Field trials on bread wheat were carried out in Marcos Juárez, Córdoba province, during 2017/18 and 2018/19 wheat growing seasons. For durum wheat, only one field trial was conducted during the 2017/18 harvest season in Necochea, Buenos Aires province. Field trial experiment locations were selected because they are the main wheat-growing areas for either bread or durum wheat. Cultivar BioINTA 1005 and ACA1901F were evaluated for bread and durum wheat, respectively; both cultivars were sown at the end of July and are susceptible to *F. graminearum*. A random block design was used for all the field trials with three replicates (plots) per treatment. The experimental plots consisted in 3 rows (2 m row^−1^, 0.2 m separation between rows, and approximately 250 heads per plot). The application of *B. velezensis* RC218, chitosan, and the pathogen was carried out at the anthesis period, corresponding to heads showing 50% of the flowering stage (Feekes 10.5.2); to treatment applications previously, heads were misted with water for 1 min to increase the relative humidity. Treatments (Table 2) were applied by using a CO_2_ commercial sprayer with three linear sprinklers (Teejet Technology) adjusted to 30 mbar to obtain a flow of 15 mL per second. In order to increase the relative humidity in the experiment, water sprinklers (fine misting) were located between plots and turned on for 5 min every 30 min from 8 am to 18 pm. Disease incidence and severity were evaluated 21 days after treatment inoculations; incidence was determined by counting spikelets showing FHB symptoms (browny, decolored) from the total spikes of the plot, while severity was visually estimated by comparison with the previously mentioned scale [50]. 

#### 4.3.2. Fusarium Head Blight Parameters at Harvested Grains: DON Content, Fusarium-Damaged Kernels, and Pathogen DNA Level

At the maturity stage, grains from each treatment (plot) were harvested and finely ground in a laboratory miller. Toxin extraction was performed with a 15 g subsample in acetonitrile:water (84:16, 100 mL) by using the clean-up Mycosep^TM^ 225 column (Romer Labs Inc., Newark, DE, USA), and the manufacturer protocol for the further determination of DON content was performed by liquid chromatography. The HPLC system consisted of a Hewlett Packard model 1100 pump (Palo Alto, CA, USA) connected to a Hewlett Packard 1100 Series variable-wavelength detector and a data module Hewlett Packard Kayak XA (HP ChemStation Rev. A.06.01, Palo Alto, CA, USA). Chromatographic separations were performed on a Luna™ C18 reversed-phase column (100 × 4.6 mm, 5 μm particle size, Phenomenex, Torrance, CA, USA) connected to a guard column SecurityGuard™ (Phenomenex, Torrance, CA, USA) (4 × 3.0 mm) filled with the same phase. The mobile phase consisted of methanol/water (12:88, v v^−1^), at a flow rate of 1.5 mL min^−1^. The detector was set at 220 nm with an attenuation of 0.01 Absorbance Units Full Scale (AUFS). The injection volume was 50 μL and the retention time of DON was 800 s. Quantification was relative to external standards of 1 to 4 μg mL^−1^ in methanol/water (5:95). The quantification limit was 0.5 μg g^−1^. 

*Fusarium*-damaged kernels were determined in order to estimate disease severity in grains, which can be used as an alternative indicator when spikes are already mature and it is impossible to use a visual scale. A total of 100 grains per treatment/plot was analyzed and FDK was defined as the amount of scabby grains (pink or white, discolored, shriveled) from the total sample. FDKs from field experiments were only analyzed in this case.

DNA content in grains was determined by TaqMan quantitative PCR. Five grams of each treatment from the durum wheat field experiment was finely pulverized in a miller (Cyclotech, Foss Tecator, Baroda, MI, USA) and 10 mg was weighed for total DNA extraction by using the DNeasy 96 plant kit (Qiagen) following the manufacturer´s instruction. qPCR amplifications were performed in 25 µL, with 12.5 µL of universal TaqMan Master Mix (Applied Biosystems, Waltham, MA, USA), 100 nM of the FAM-labeled probe and internal control probe, and 400 nM of forward and reverse primers for *F. graminearum ss* detection [52]. Amplifications were performed in an ABI Prism 7500 Sequence Detection System (Applied Biosystems). Cycling conditions were a single cycle of 2 min to 50 °C to degrade uracil containing DNA and 10 min to 95 °C to inactivate uracil-N-glicosidase, followed by 40 cycles of 95 °C for 15 s and 60 °C for 1 min. A standard curve was generated by using tenfold serial dilutions of pure *F. graminearum* DNA in the range of 0.1 to 10^4^ pg µL^−1^, and five replicates of the pathogen quantifications were carried out.

### 4.4. Statistical Analyses

FHB disease severity data on greenhouse experiments were subjected to ANOVA and means were separated by Duncan’s method (*p* ≤ 0.05). Field experiment data and *Fusarium*-damaged kernels were subjected to ANOVA and means were separated by Fisher’s LSD method (*p* ≤ 0.05). 

## Figures and Tables

**Figure 1 toxins-14-00499-f001:**
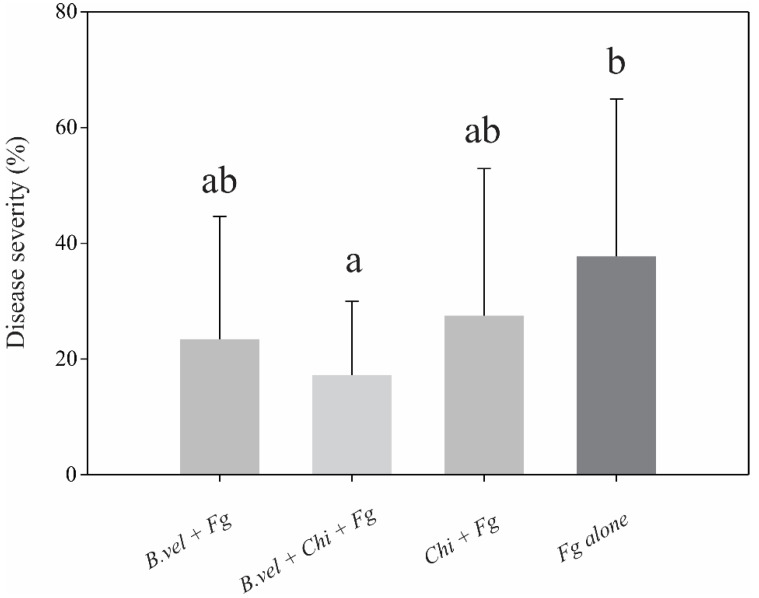
FHB evaluation on greenhouse trial during 2016 on a susceptible bread wheat cultivar. On *x* axis, treatments are: *B. vel*, *B. velezensis* RC218 applied at 10^6^ ufc mL^−1^; Fg, *Fusarium graminearum ss* applied at 10^5^ conidia mL^−1^; *Chi*, chitosan 0.1%. Different letters on the columns indicate significant differences according to Duncan´s test (*p* ≤ 0.05).

**Figure 2 toxins-14-00499-f002:**
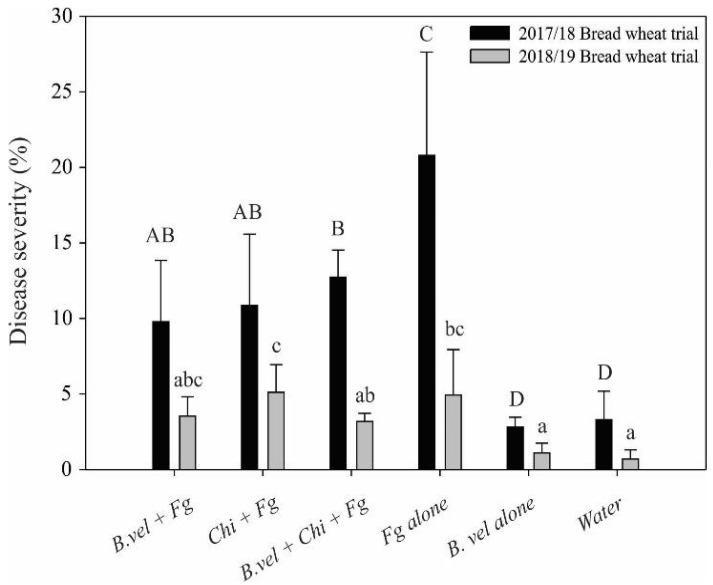
FHB severity observed during 2017/18 and 2018/19 field trials conducted at Marcos Juárez on bread wheat. Disease severity means were separated by Fisher’s LSD method (*p* ≤ 0.05) for either 2017/18 (Capital) or 2018/19 (lower case) wheat trial. Bars with different letters indicate significant differences. Treatments are: *B. vel*, *B. velezensis* RC218 applied at 10^6^ ufc mL^−1^; Fg, *Fusarium graminearum ss* applied at 10^5^ conidia mL^−1^; *Chi*, chitosan 0.1%; *Water*, negative control.

**Figure 3 toxins-14-00499-f003:**
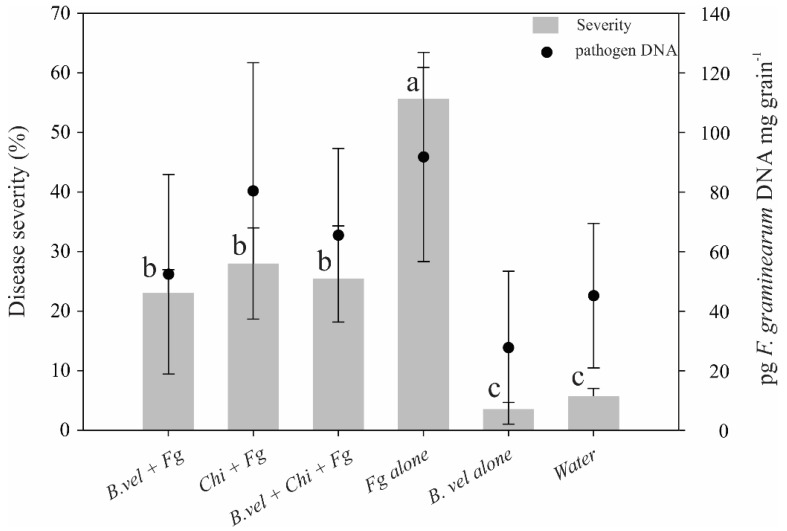
FHB disease severity evaluated on durum wheat in 2017/18 season at Necochea. Disease severity means were separated by Fisher’s LSD method (*p* ≤ 0.05). Columns with different letters indicate significant differences. Treatments are: *B. vel*, *B. velezensis* RC218 applied at 10^6^ ufc mL^−1^; Fg, *Fusarium graminearum ss* applied at 10^5^ conidia mL^−1^; *Chi*, chitosan 0.1%; *Water*, negative control.

**Table 1 toxins-14-00499-t001:** Deoxynivalenol accumulation and *Fusarium*-damaged kernels in bread and durum wheat trials.

Treatments	Deoxynivalenol (µg g^−1^) *		FDK	
	Bread	Durum	Bread	Durum
*B. velezensis* RC218 + *F. graminearum ss* (*n* = 3)	2.20 ± 0.50 b	2.60 ± 2,30 a	2.7 ± 1.50 ab	7.3 ± 1.40 bc
Chitosan (0.1%) + *F. graminearum ss* (*n* = 3)	1.42 ± 0.87 b	1.80 ± 1,70 a	4 ± 2.60 ab	6.8 ± 0.70 ab
*B. velezensis* RC218 + Chitosan + *F. graminearum ss* (*n* = 3)	1.57 ± 0.22 b	1.80 ± 1.60 a	2 ± 1.70 a	8.3 ± 3.70 e
*F. graminearum ss* alone (*n* = 3)	4.43 ± 1.70 a	6.12 ± 1.50 b	5 ± 20 b	11.2 ± 3.50 e
Water control (*n* = 3)	0.30 ± 0.13 c	Nd	1.30 ± 0.50 a	6.40 ± 0.30 a
*B. velezensis* RC218 control (*n* = 3)	Nd	Nd	1.70 ± 0.5 a	8 ± 2.30 d

* Mean of deoxynivalenol levels and FDK for bread or durum wheat were separated by Fisher’s LSD method (*p* ≤ 0.05). On each column, different letters indicate significant differences. FDK: *Fusarium*-damaged kernels. Nd: not detected, detection limit 50 µg Kg^−1^.

**Table 2 toxins-14-00499-t002:** Treatments description for 2016 greenhouse experiment and field trials.

Treatments	Concentration Applied *
*B. velezensis* RC218 + *F. graminearum ss*	10^6^ + 10^5^
Chitosan + *F. graminearum*	0.1% + 10^5^
*B. velezensis* RC218 + chitosan + *F. graminearum ss*	10^6^ + 0.1% + 10^5^
Possitive control (*F. graminearum ss* alone)	10^5^
Negative control (water)	-

* A total of 10 mL was applied on each pot from each treatment. *B. velezensis* RC218 at a concentration of 1 × 10^6^ cfu mL^−1^; *F. graminearum ss* at 1 × 10^5^ conidia mL^−1^ and chitosan at 0.1%. In all involved *F. graminearum ss* treatments, the pathogen was applied for 30 min after either *B. velezensis* RC218 or chitosan.

## Data Availability

Not applicable.

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
