# Peer review of "Combination of Bacillus velezensis RC218 and Chitosan to Control Fusarium Head Blight on Bread and Durum Wheat under Greenhouse and Field Conditions"

_toxins, 2022, doi:10.3390/toxins14070499_

Round 1

Reviewer 1 Report

The manuscript reports a nice extension of previous studies (e.g., Palazzini et al., Microbio Res, 2016, 192, 30-36; Palazzini et al., Toxins 2018, 10, 88). There has been interests in applying Bacillus velezensis RC 218 as a biocontrol agent against FHB and mycotoxin production, so in this study the authors demonstrated some combined effects of Bacillus velezensis RC 218 and Chitosan under various conditions.  Below please find a few minor comments for consideration.

Figure 2. There are obvious differences between 2017/2018 and 2081/2019 trials. It would be helpful to further explain the trends and differences between the two trials in the discussion section.

T    Table 1. In terms of DON concentrations, the deduction in Durum is not as significant as in other crops. Why?

3    Table 1. Why was DON detected in the negative controls (water control) for Bread and Durum? A separate section should be included in the Materials and Methods to document how controls were prepared for the DON determination.

     Table 1. If standard deviations reported, include the number of replicates (n=?)

I    In figures and tables, it is hard to follow the letters used to show which pair was compared and which pair is significantly different. Relabel the figures and tables or revise table/figure notes to clarify.

Author Response

Reviewer 1:

The manuscript reports a nice extension of previous studies (e.g., Palazzini et al., Microbio Res, 2016, 192, 30-36; Palazzini et al., Toxins 2018, 10, 88). There has been interests in applying Bacillus velezensis RC 218 as a biocontrol agent against FHB and mycotoxin production, so in this study the authors demonstrated some combined effects of Bacillus velezensis RC 218 and Chitosan under various conditions.  Below please find a few minor comments for consideration.

Figure 2. There are obvious differences between 2017/2018 and 2081/2019 trials. It would be helpful to further explain the trends and differences between the two trials in the discussion section.

Response: A paragraph was added in lines 318-321 in order to discuss a possible explanation of the differences observed between the two harvest seasons .

     Table 1. In terms of DON concentrations, the deduction in Durum is not as significant as in other crops. Why?

Response: From your question we understand why it was the difference between durum and bread wheat in the reduction in DON content ? In that case, the table did  not show a reduction percentage (%) because it was too much to add in the table. Nevertheless, reduction in DON content for durum wheat was higher (more than 50%) than for bread wheat. To our knowledge, durum wheat in more susceptible to FHB and DON accumulation than bread wheat.

    Table 1. Why was DON detected in the negative controls (water control) for Bread and Durum? A separate section should be included in the Materials and Methods to document how controls were prepared for the DON determination.

    Response:  The negative control (Water control in our table) is subjected to potential natural infection given by F. graminearum ss already present in the field. This treatment was processed as the other treatments: harvest of grains, milling and DON extraction by Romer 225 and HPLC determination as described in M&M section. All treatments

     Table 1. If standard deviations reported, include the number of replicates (n=?)

Response: The number of replicates were added according to the reviewer´s comment .

I    In figures and tables, it is hard to follow the letters used to show which pair was compared and which pair is significantly different. Relabel the figures and tables or revise table/figure notes to clarify.

Response: Table 1: the footnote was rewritten. We hope that now is more clear.

Reviewer 2 Report

Authors described the evaluation of a combination of chitosan with the known biocontrol agent in relation to the control of the FHB disease as well as DON contamination in bread and durum wheat. The problem of FHB and DON contamination of wheat is relevant for many regions of the world, as well as the studies intended to find an efficient biocontrol method to reduce the use of chemical fungicides. From this point of view, the choice of this theme of study is quite substantiated. Authors describe greenhouse and field experiments that is the benefit of this manuscript since the data obtained in vitro often differ from those obtained under field conditions. The results obtained by authors are not too inspiring, though demonstrate some improvement in relation to the resulting levels of disease severity or mycotoxin contamination of grain. At the same time, the discussion part is rather weak and almost do not involve the obtained results. In my opinion, it should be re-written prior to accept this paper for publication. I also have some minor comments to other manuscript parts (see below) as well as recommend a moderate language editing.

Introduction

  1. Authors provide a good description of the current situation with the bio-methods to control Fusarium fungi. They also substantiated the choice of chitosan as a compound able to reduce FHB severity and DON accumulation. It would be also good to add some information about the choice of a strain they used in the study. Why do you use this B. velezensis strain? Was this choice based on some preliminary evaluation of its activity towards Fusarium fungi and DON?
  2. The introduction seems to have an excess number of references (41 out of 70). Probably it would be good to remove the excess information from this part of the manuscript. For example, authors can shorten the text between the lines 45-54; there is no need to list all these maximum permissible DON levels. There is also no need to mention so much references in lines 70-73; I also consider the part of introduction describing the benefits of chitosan can be also reduced without any significant losses of the paper quality.

Results

Line 106: which former strain do you mean? The phrase is constructed so that it seems you speak about any other B. velezensis strain. If you mean RC218, the world “former” is unnecessary, you can simply delete it.

Line 134: please, move Table 1 close to this paragraph, i.e. before Fig. 3.

Line 134: “… combined treatment was the most successful” - only for bread wheat, since for durum wheat FDK values for Fgra and combined treatment were similar (e-group). Please, make the corresponding correction or remove “durum” from line 117.

Line 149-150: if you do not show data on the disease incidence for bread wheat and do not show such data for durum wheat, then why do you mention it at all?

Line 154: why did you determine Fg DNA content only for durum wheat, but not for bread wheat?

Discussion

This section of the manuscript is rather weak. It contains mainly the listing of results obtained in other studies in relation to chitosan or some biocontrol bacteria. At the same time, authors almost do not discuss their results and the related prospects. Actually, they do two-season studies of bread and durum wheat with rather ambiguous results. For some cases/parameters, combination of chitosan and Bacillus strain showed better results than those components used alone, but in other cases no significant difference was observed. Of course, efficiency of any biocontrol approaches in the field strongly depends on the season and weather and other environmental conditions, so sometimes it is rather low and has not any economical benefits. Nevertheless, based on the presented results, it is difficult to conclude, whether the combined use of chitosan and the biocontrol strain has some benefits. In my opinion, this point should be discussed. In addition, such studies are often arranged using a chemical fungicide as a “treated control” to get readers a possibility to evaluate the relative efficiency of the biocontrol method. Without this information, the valuability of this study is significantly reduced. I do not demand the re-arrangement of experiments with inclusion of a chemical treatment variant (since it will require two or three additional seasons), but authors at least have to discuss the potential of the tested combination in view of the obtained results as well as in comparison with the results obtained by other authors with other biocontrol agents and results of chemical protection, so a reader could better understand the (possible) profitability of the proposed combination in practical use.

I would also recommended authors to add the evaluation of the combined effect of the tested components from the point of view of synergistic or additive interaction as it may be easily calculated using existing formula. If the synergism or additivity of the effect will be confirmed by such calculations, this will be an additional evidence of the prospects for use of such combined preparations.

In addition, the first three paragraphs of discussion repeat some information given in the Introduction. It seems there is no need to do so. You’ve already described the drawbacks of chemical control and benefits of the biocontrol.

Materials and methods

Line 269: which properties of RC218 were evaluated in the earlier studies? I consider this information would be more suitable for the Introduction as it substantiates the choice of this strain for the study.

Line 281: which laboratory?

Line 281-283: what is the purpose of these references (10 and 67)? Please, explain.

Lines 311-312: if you give the sowing date for one cultivar, then it would be good to do the same for another one.

Line 315: 2 m row-1 - did you mean the row length was 2 m?

Author Response

Reviewer 2:

Authors described the evaluation of a combination of chitosan with the known biocontrol agent in relation to the control of the FHB disease as well as DON contamination in bread and durum wheat. The problem of FHB and DON contamination of wheat is relevant for many regions of the world, as well as the studies intended to find an efficient biocontrol method to reduce the use of chemical fungicides. From this point of view, the choice of this theme of study is quite substantiated. Authors describe greenhouse and field experiments that is the benefit of this manuscript since the data obtained in vitro often differ from those obtained under field conditions. The results obtained by authors are not too inspiring, though demonstrate some improvement in relation to the resulting levels of disease severity or mycotoxin contamination of grain. At the same time, the discussion part is rather weak and almost do not involve the obtained results. In my opinion, it should be re-written prior to accept this paper for publication. I also have some minor comments to other manuscript parts (see below) as well as recommend a moderate language editing.

Introduction

  1. Authors provide a good description of the current situation with the bio-methods to control Fusarium fungi. They also substantiated the choice of chitosan as a compound able to reduce FHB severity and DON accumulation. It would be also good to add some information about the choice of a strain they used in the study. Why do you use this B. velezensis strain? Was this choice based on some preliminary evaluation of its activity towards Fusarium fungi and DON?

Response: The paragraph was rewritten (lines 87-92) in order to strength the selection of B. velezensis RC218 as the best candidate to be evaluated in combination with chitosan.

The introduction seems to have an excess number of references (41 out of 70). Probably it would be good to remove the excess information from this part of the manuscript. For example, authors can shorten the text between the lines 45-54; there is no need to list all these maximum permissible DON levels. There is also no need to mention so much references in lines 70-73; I also consider the part of introduction describing the benefits of chitosan can be also reduced without any significant losses of the paper quality.

Response: According to the reviewer´s  suggestion  several references in this paragraph were removed since they were cited only once. Chitosan description was  also shortened

Results

Line 106: which former strain do you mean? The phrase is constructed so that it seems you speak about any other B. velezensis strain. If you mean RC218, the world “former” is unnecessary, you can simply delete it.

Response: Correct, the word “former” was deleted.

Line 134: please, move Table 1 close to this paragraph, i.e. before Fig. 3.

Response: Table 1 was moved before figure 2 since it was first described in the text.

Line 134: “… combined treatment was the most successful” - only for bread wheat, since for durum wheat FDK values for Fgra and combined treatment were similar (e-group). Please, make the corresponding correction or remove “durum” from line 117.

Response: According to the reviewer´s comment the line 116 was corrected.

Line 149-150: if you do not show data on the disease incidence for bread wheat and do not show such data for durum wheat, then why do you mention it at all?

Response: Incidence is an important parameter for the evaluation of several fungal diseases. The problem we have observed in previous studies is that this parameter not always followed the same trend when applying biocontrol treatments, but normally is difficult to reduce this parameter, but not for disease severity. Year by year, we achieved reductions in DON, disease severity and FDK, but not on the incidence parameter.

Line 154: why did you determine Fg DNA content only for durum wheat, but not for bread wheat?

Response: In this study, we only determined Fg DNA only in durum wheat since in previous studies we have observed good correlations between DNA and disease severity in bread wheat (Palazzini et al., 2015). Additionally, a study on durum wheat was carried out during the previous harvest season  (2016/17) (Palazzini et al., 2018) were F. graminearum ss  DNA was measured in combination with B. velezensis RC218 but not  in the chitosan treatment. We considered important to evaluate pathogen DNA under the treatments evaluated in the present study in  durum wheat.

Publications:

Cereal Research Communications 2015, DOI: 10.1556/0806.43.2015.017

Toxins 2018, 10, 88; doi:10.3390/toxins10020088

Discussion

The section was rewritten and reorganized according to the reviewer´s suggestion, some aspects are clarified below

This section of the manuscript is rather weak. It contains mainly the listing of results obtained in other studies in relation to chitosan or some biocontrol bacteria. At the same time, authors almost do not discuss their results and the related prospects. Actually, they do two-season studies of bread and durum wheat with rather ambiguous results. For some cases/parameters, combination of chitosan and Bacillus strain showed better results than those components used alone, but in other cases no significant difference was observed. Of course, efficiency of any biocontrol approaches in the field strongly depends on the season and weather and other environmental conditions, so sometimes it is rather low and has not any economical benefits. Nevertheless, based on the presented results, it is difficult to conclude, whether the combined use of chitosan and the biocontrol strain has some benefits. In my opinion, this point should be discussed.

Response: The results obtained with the combination of the treatments were further discussed in this section.

In addition, such studies are often arranged using a chemical fungicide as a “treated control” to get readers a possibility to evaluate the relative efficiency of the biocontrol method. Without this information, the valuability of this study is significantly reduced. I do not demand the re-arrangement of experiments with inclusion of a chemical treatment variant (since it will require two or three additional seasons), but authors at least have to discuss the potential of the tested combination in view of the obtained results as well as in comparison with the results obtained by other authors with other biocontrol agents and results of chemical protection, so a reader could better understand the (possible) profitability of the proposed combination in practical use.

Response: We know that chemical control is effective under different conditions. Triazole tolerance by B. velezensis RC218 was evaluated previously (Palazzini et al., 2018) in order to use the biocontrol agent under an integrated pest management. In this study, we did not considered to compare biocontrol, chitosan and a chemical control; this kind of studies  are ongoing by using reduced doses of chemicals in combination with biocontrol versus the doses recommended by the fungicide supplier

Letters in Applied Microbiology 66, 434-438; doi:10.1111/lam.12869

I would also recommended authors to add the evaluation of the combined effect of the tested components from the point of view of synergistic or additive interaction as it may be easily calculated using existing formula. If the synergism or additivity of the effect will be confirmed by such calculations, this will be an additional evidence of the prospects for use of such combined preparations.

Response: According to the reviewer´s suggestion, calculation of additive or synergistic effect was added to the manuscript and analyzed. Description of the methodology used is included in lines 431-438 and in results section is also presented for the greenhouse trial (line 117). Also, the proper reference was added in the material and methods section and in the reference list.

 In addition, the first three paragraphs of discussion repeat some information given in the Introduction. It seems there is no need to do so. You’ve already described the drawbacks of chemical control and benefits of the biocontrol.

Response: The three first paragraphs were shortened to only one, according to the reviewer suggestion.

Materials and methods

Line 269: which properties of RC218 were evaluated in the earlier studies? I consider this information would be more suitable for the Introduction as it substantiates the choice of this strain for the study.

Response: The information was added in the introduction section on lines 87-92, giving more strength to the choice of the strain.

Line 281: which laboratory?

Response: The sentence was rewritten in order to clarify the origin of the Fusarium graminearum ss  strains  used in the experiments.

Line 281-283: what is the purpose of these references (10 and 67)? Please, explain.

Response: The references cited were removed.

Lines 311-312: if you give the sowing date for one cultivar, then it would be good to do the same for another one.

Response: The sentence was rewritten as you suggested.

Line 315: 2 m row-1 - did you mean the row length was 2 m?

Response: Yes, the reviewer is correct.

Reviewer 3 Report

The manuscript is devoted to the search for an ecological means of protecting grain crops from Fusarium graminearum sensu stricto, which would not only increase the yield, but also reduce fungal toxins in grains. From the existing approaches to solving this problem, the authors chose the method of using a biological agent that suppresses the growth of Fusarium, namely the strain of Bacillus velezensis RC218. In addition to the indicated bacillus strain, wheat was simultaneously treated with chitosan. This combination of two antifungal agents was performed for the first time. The objects were bread and durum wheat. The studies were carried out both in the greenhouse and in the field for two years. A decrease in the severity of plant diseases was noted with such a double treatment.

The work was performed at a good adequate modern methodological level, statistical processing was carried out. The manuscript contains two tables and three figures. References to literature on this subject are legitimate.

The article does not require corrections and can be published in its original form.

Author Response

Dear Reviewer, thanks for your comments.

Round 2

Reviewer 2 Report

GENERAL

Authors made some work to improve the manuscript and addressed my comments.

In the case of greenhouse experiments, the performed evaluation of a possible additivity/synergism of the effects of two tested components towards reduction of the disease severity, chitosan and Bvel strain showed lack of any additive or synergistic interaction, i.e., combination of these two components does not result in a significant improvement of the total effect compared to the effects of these compounds taken alone. Though authors did not do such calculations for other described experiments, it seems they may also show lack of additive or synergistic effect. Thus, such combination of biocontrol agents seems to have no practical advantages. Authors consider it can be caused by insufficient chitosan concentration and say the further research is needed to optimize it for field treatments. The lack of any benefits from a combined treatment with two compounds is a quite often phenomenon. An increase of the chitosan concentration may change the situation, but in the case of too high resulting effect, there can be problems with the calculation of a possible synergism/additivity since it is difficult to calculate it at highly effective concentrations (even if presents).

At this moment, the main result of this study is: though chitosan and Bvel show good results towards FHB inhibition and DON production, especially in durum wheat, their combination does not significantly improve the effect; no synergistic and even additive effect was shown. Along with several minor comments, there is one major question concerning the proper use of a formula for calculation of synergistic/additive effect (see below). Probably, after a re-calculation, the results will be more inspiring. If not, then the question arises: are the obtained results (i.e., lack of any improvement in the disease suppression effect in the case of combined use of two agents) have any value to be published in the Q1 journal? What new and valuable information authors want to tell other scientists? As I understand, the positive effects of the Bvel strain and chitosan on the studied disease have been described in the previous publications, so the only new thing is that authors tested combination of these two factors. If they would obtain good results, i.e., an obvious synergistic or even additive effect, it would make sense to publish them as they could be used for the further practical and probably theoretical studies. In the absence of such effects, these results have no significance for other scientists. They mean only that there is no sense to use this strain and this chitosan type together. This is rather a particular case, and the situation can be quite different with other strains or other types of chitosan. Almost each lab gets a lot of similar results during its work, but does not publish it. Therefore, I strongly recommend authors to think and try to substantiate the value of their results and accentuate them in the discussion. If not, then I would not recommend this manuscript for publication in a Q1-rank journal in the current form. At the same time, these results can be used together with some other, more valuable data. For example, if the use of higher chitosan concentration will result in the desired effect, or the current chitosan concentration will work in the additive or synergistic manner with another Bvel strain, or the used combination will provide a significant effect towards another pathogen strain, then the paper may combine these and new data…

MINOR COMMENTS

Line 108: “…resulted in a statistically reduction…”: statistically SIGNIFICANT?

Line 140: I did not find any explanation of the FDK abbreviation above this table. Please, give an explanation in the note below the table. In addition, since Fig 2 is mentioned prior Table 1 (both in the same paragraph), Table 1 should be placed after Fig. 2. In addition, the data shown in the table should be presented in the same manner concerning the number of ciphers after a decimal point, i.e., not 2.2 and 1.42, but either 2.20 and 1.42, or 2.2 and 1.4. Please, correct.

Line 215: did not render (not rendered).

Line 217-218: “…since the inoculum dose used for B. velezensis RC218 has been previously demonstrated.” Demonstrated what?

Line 264-265: …several harvest seasonS.

Line 277: please, add information about the location (city, country) of the institute.

MAIN COMMENT

Lines 297-303:

1. The most traditional way to determine a possible synergism is the use of a Limpel’s formula (see Richer D.L. (1987). Pesticide Sci., 19(4), 309–315. doi:10.1002/ps.2780190408).

It looks like E = X + Y - XY/100 (*),

where X and Y represent inhibition of a pathogen/weed growth by each of the compounds (%) and E is the expected growth inhibition by a combination of both compounds (%).

If E < Ereal, i.e. real effect obtained in the experiment, then the synergism is confirmed.

The original Boyette paper cited in your manuscript and describing how to analyze treatment interactions has a reference to the paper published by Colby (Colby S.R. (1967). Weeds, 15(1), 20. doi:10.2307/4041058). In the last paper, the Limpel’s formula was transformed to the “inverted” formula of the following form:

E1 = X1Y1/100 (**),

where X1 and Y1 represent the pathogen/weed growth in the presence of each tested compound (% to the control) and E1 is the expected growth in the presence of both compounds. In this case, the synergism is considered to be confirmed if E1 > Ereal.

Note that this formula operates with the growth/survival parameters, not growth inhibition or disease reduction. In your calculations, you used formula (**) but X and Y represented disease reduction that is completely wrong. In your case you should either use formula (*) with the disease reduction values, or formula (**) with disease severity values in % of the control. Also, the corresponding evaluation criterion for E/E1 and Ereal should be used. Please, check and correct.

2. Describing the calculation of a possible additive effect via calculation of the R ratio, Boyette et al. cited Gisi et al. (Gisi et al. (1985) Transact. British Mycol. Soc., 85(2), 299–306. doi:10.1016/s0007-1536(85)80192-3). However, if you will look on this paper, you will see that Gisi operates with EC90 values in the formula R = EC90(exp)/EC90(observed), i.e., the formula was proposed for use with concentrations of tested compounds causing 90% response. Therefore, this formula seems to be not applicable for your case (and for the Boyette case too). Please, check thoroughly.

In this situation, I would recommend authors to re-calculate the results using the Limpel’s formula for disease suppression levels; in this case, E = Ereal actually corresponds to the additive effect. See, for example, Shcherbakova et al. (2021) Front. Microbiol. 12,      https://doi.org/10.3389/fmicb.2021.629429

Author Response

Answers to Reviewer 2 Round 2:

Line 108: “…resulted in a statistically reduction…”: statistically SIGNIFICANT?

Response: the word “significant” was added to the sentence according to the reviewer´s suggestion.

Line 140: I did not find any explanation of the FDK abbreviation above this table. Please, give an explanation in the note below the table. In addition, since Fig 2 is mentioned prior Table 1 (both in the same paragraph), Table 1 should be placed after Fig. 2. In addition, the data shown in the table should be presented in the same manner concerning the number of ciphers after a decimal point, i.e., not 2.2 and 1.42, but either 2.20 and 1.42, or 2.2 and 1.4. Please, correct.

Response: according to your suggestions, FDK description was added in the table and it was also moved below Fig. 2. Data in the table was normalized in relation to ciphers after the decimal point.

Line 215: did not render (not rendered).

Response: The word was corrected as the reviewer suggested.

Line 217-218: “…since the inoculum dose used for B. velezensis RC218 has been previously demonstrated.” Demonstrated what?

Response: the sentence was modified in order to give strength to the dose used in B. velezensis RC218 under field trials.

Line 264-265: …several harvest seasonS.

Response: The word was corrected according to your suggestion.

Line 277: please, add information about the location (city, country) of the institute.

Response: The sentence was modified by the addition of “(Río Cuarto city, Argentina)”, according to your suggestion.

MAIN COMMENT

Lines 297-303:

  1. The most traditional way to determine a possible synergism is the use of a Limpel’s formula (see Richer D.L. (1987). Pesticide Sci., 19(4), 309–315. doi:10.1002/ps.2780190408).

It looks like E = X + Y - XY/100 (*),

where X and Y represent inhibition of a pathogen/weed growth by each of the compounds (%) and E is the expected growth inhibition by a combination of both compounds (%).

If E < Ereal, i.e. real effect obtained in the experiment, then the synergism is confirmed.

The original Boyette paper cited in your manuscript and describing how to analyze treatment interactions has a reference to the paper published by Colby (Colby S.R. (1967). Weeds, 15(1), 20. doi:10.2307/4041058). In the last paper, the Limpel’s formula was transformed to the “inverted” formula of the following form:

E1 = X1Y1/100 (**),

where X1 and Y1 represent the pathogen/weed growth in the presence of each tested compound (% to the control) and E1 is the expected growth in the presence of both compounds. In this case, the synergism is considered to be confirmed if E1 > Ereal.

Note that this formula operates with the growth/survival parameters, not growth inhibition or disease reduction. In your calculations, you used formula (**) but X and Y represented disease reduction that is completely wrong. In your case you should either use formula (*) with the disease reduction values, or formula (**) with disease severity values in % of the control. Also, the corresponding evaluation criterion for E/E1 and Ereal should be used. Please, check and correct.

  1. Describing the calculation of a possible additive effect via calculation of the R ratio, Boyette et al. cited Gisi et al. (Gisi et al. (1985) Transact. British Mycol. Soc., 85(2), 299–306. doi:10.1016/s0007-1536(85)80192-3). However, if you will look on this paper, you will see that Gisi operates with EC90 values in the formula R = EC90(exp)/EC90(observed), i.e., the formula was proposed for use with concentrations of tested compounds causing 90% response. Therefore, this formula seems to be not applicable for your case (and for the Boyette case too). Please, check thoroughly.

In this situation, I would recommend authors to re-calculate the results using the Limpel’s formula for disease suppression levels; in this case, E = Ereal actually corresponds to the additive effect. See, for example, Shcherbakova et al. (2021) Front. Microbiol. 12,      https://doi.org/10.3389/fmicb.2021.629429

Response: We strongly appreciate the reviewer´s suggestions in describing and explaining the diverse equations and formulas used to analyze the effects of combining two active principles. We are sorry to use firstly the Boyette (and Colby formula) in the wrong sense. Based on your suggestions, Limpel´s formula (*) was used to calculate the possible synergism or additive effect between treatments. The calculated Ee was 54.64 meanwhile the real Er was 54.11. As the obtained results were quite similar (Ee vs Er), we concluded that only an “additive” effect was observed for the combination of Bvel and chitosan at the doses evaluated. Based on this, the results, discussion and material sections were modified (Lines 171, 302-304, 625-627, 726-733).

Limpel Calculation for greenhouse Disease severity
  Trat Bvel trat Chi Bvel+Chi Fgram
Greenhouse 23,53 27,40 17,30 37,70

  Disease Reduction %  
  Red Bvel Red Chi Red Combined  
  37,59 27,32 54,11  

  Bv+Chi indiv BvXChi/100 E calculated E real
  64,91 10,27 54,64 54,11
